# Impact of SARS-CoV-2 Infection in Patients with Neurological Pathology

**DOI:** 10.3390/diagnostics12020473

**Published:** 2022-02-12

**Authors:** Any Docu Axelerad, Lavinia Florenta Muja, Alina Zorina Stroe, Lavinia Alexandra Zlotea, Carmen Adella Sirbu, Silviu Docu Axelerad, Dragos Catalin Jianu, Corina Elena Frecus, Cristina Maria Mihai

**Affiliations:** 1Department of Neurology, General Medicine Faculty, Ovidius University, 900470 Constanta, Romania; axelerad.docu@365.univ-ovidius.ro (A.D.A.); lavinia.muja@365.univ-ovidius.ro (L.F.M.); 2Radiology-Medical Imaging, Sf. Apostol Andrei Emergency County Clinical Hospital Constanta, 900591 Constanta, Romania; laviniazlotea23@gmail.com; 3Department of Neurology, Titu Maiorescu University, 040441 Bucharest, Romania; carmen.sirbu@prof.utm.ro; 4Faculty of General Medicine, ‘Vasile Goldis’ University, 317046 Arad, Romania; docu-axelerad.silviu@student.uvvg.ro; 5Department of Neurology, Victor Babes University of Medicine and Pharmacy, 300041 Timisoara, Romania; jianu.dragos@umft.ro; 6Department of Pediatrics, General Medicine Faculty, Ovidius University, 900470 Constanta, Romania; corina.frecus@365.univ-ovidius.ro (C.E.F.); cristina_mihai@365.univ-ovidius.ro (C.M.M.)

**Keywords:** COVID-19, SARS-CoV-2 infection, neurological pathology, imaging investigations

## Abstract

The COVID-19 disease, caused by infection with SARS-CoV-2, rapidly transformed into a pandemic following its emergence, and it continues to affect the population at a global level. This disease is associated with high mortality rates and mainly affects the pulmonary spectrum, with signs of interstitial pneumonia or other pathological modifications. Signs indicative of SARS-CoV-2 infection can be observed using pulmonary radiography or computed tomography scans and are closely linked to acute respiratory distress; however, there is accumulating evidence that the virus affects the central nervous system. Several symptoms, such as headaches, cough, fatigue, myalgia, ageusia, and anosmia, have also been reported along with neurological syndromes such as stroke, encephalopathy, Guillain–Barre syndrome, convulsions, and coma; the most frequent associated complication is ischemic stroke. Diagnosis of infection with SARS-CoV-2 virus is based on a positive RT-PCR test. Imaging investigations, such as thoracic computed tomography scans, are not used to diagnose COVID-19, monitor for pulmonary disease, or follow dynamic disease evolution, but they may be used in the case of a negative RT-PCR test. This paper presents the research performed on a group of 150 cases of patients affected by neurological disorders and that were subsequently confirmed to be infected with SARS-CoV-2, which was carried out over a period of 10 months within the Neurology Department and Laboratory of Magnetic Resonance Imaging of “Sf. Andrei” Emergency Hospital in Constanta. The collected data are observational and provide perspectives on the neurological pathology associated with the SARS-CoV-2 virus, and on the frequently associated risk factors, associated comorbidities, and the ages of patients who were affected by the virus, as well as the clinical and paraclinical manifestations of the patients admitted to the hospital’s neurology department.

## 1. Introduction

The source of COVID-19, officially designated by the World Health Organization, is infection with the SARS-CoV-2 virus (severe acute respiratory syndrome coronavirus 2) [1].

Infection with the SARS-CoV-2 virus can result in a respiratory disease that mainly attacks the lungs, provoking symptoms ranging in severity from slight or moderate to severe, including fever, headaches, fatigue, dry cough, myalgia, and diarrhea. However, recent clinical studies have indicated that this type of pathogen has a vast infectious capacity to spread into the extrapulmonary tissues, provoking multi-organ insufficiency in patients with severe impairment. The capacity of the SARS-CoV-2 infection to invade the central nervous system and the peripheral nervous system currently represents a major concern [2].

Recent studies have reported certain neurological pathologies associated with SARS-CoV-2 virus infection. These studies have classified certain manifestations connected to the central nervous system, such as headaches, vertigo, impaired consciousness, acute cerebrovascular disease, and epilepsy, in addition to other symptoms related to the peripheral nervous system, such as hyposmia, anosmia, hypogeusia, ageusia, myalgia, and Guillain-Barre syndrome [3]. The most frequent neurological complication that has appeared in the SARS-CoV-2 infection is ischemic stroke [4].

A thoracic computed tomography (CT) scan is recommended for cases of suspected or confirmed SARS-CoV-2 infection, due to its association with major lung disease, with the aim of evaluating and following its progression. Compared to the thoracic CT-scanning, thoracic radiography (X-rays) is not as reliable, and cannot be used to identify disease symptoms in the early stages, but in moderate to advanced phases, it may indicate the evolution of acute respiratory distress. Moreover, real-time polymerase chain reaction (RT-PCR) tests are used to diagnose infection with the SARS-CoV-2 virus. Imaging investigations based on CT-scanning have proven useful for diagnosis, even if an initial RT-PCR test was falsely negative [5].

The study was performed within a 10-month period, between 1 October 2020, and 31 August 2021, within the Neurology Department and Laboratory of Magnetic Resonance Imaging of “Sf. Andrei” Emergency Hospital in Constanta. The Constanta County Emergency Clinical Hospital’s Ethics Committee for Clinical Studies (registration number 34/26.11.2021) authorized this study, which followed the guidelines of the Declaration of Helsinki. Prior to enrollment, all individuals provided written informed consent.

Following subjective and objective clinical examinations, and based on paraclinical investigations, the study group consisted of 150 cases of patients admitted to the neurology department of the hospital with various neurological diseases and who were declared to be positively infected with the virus according to the presence of ribonucleic acid (RNA) SARS-CoV-2, as determined by RT-PCR test.

## 2. Results

The study was performed over a period of 10 months, between 1 October 2020, and 31 August 2021. The total number of cases studied was 150. In terms of the patients’ sex, the number of cases was almost equal, with 76 cases of men and 74 cases of women.

### 2.1. Type of Neurological Disease

Table 1 shows the types and number of cases of neurological diseases presented by patients admitted to the neurology department.

### 2.2. Submission of Clinical and Imaging Analysis of Representative Cases of the Studied Group

#### 2.2.1. Case 1

A 49-year-old patient presented herself to the emergency room for ascending lower-limb paresthesia and lower-limb motor deficit, with denial of any infectious episode. Upon neurological assessment at admission, the patient was conscious, cooperative, temporo-spatially oriented, presented no rolling of the neck, normal oculomotor coordination, no nystagmus, paraparesis 3–4/5, had lively osteotendinous reflexes, no bilateral strength deficit in dorsal and plantar flexion, normal deep myo-arthro-kinetic sensitivity, tactile sensitivity with T9 sensitivity level, bilateral plantar skin reflex in flexion, bilateral postural tremor in the upper limbs, and was in a state of intermittent urine retention. CT images of the brain upon admission revealed an age-appropriate appearance.

On the first day of hospitalization, when tested for SARS-CoV-2 RNA by RT-PCR, the result was positive. On the same day of hospitalization, the patient underwent a lumbar puncture (element count 155/mm^3^, albumin 310 mg/L, chlorine 128 mmol/L, and glucose 70 mg/dL) and a thoracolumbar spine MRI, which revealed thoracic intramedullary lesions suggestive of an inflammatory-infectious substrate (Figure 1).

The patient was admitted for 1 day to the neurology ward. Being a patient with confirmed SARS-CoV-2 infection, she would ordinarily be transferred to a COVID-19-supported hospital, but she refused, and contrary to medical advice, was discharged on request. During hospitalization, the patient received hydroelectrolytic rebalancing treatment, vitamin therapy, cortico-therapy, and gastric protection.

The neurological assessment at discharge found the patient to be conscious and cooperative, with no neck roll, no nystagmus, paraparesis 3/5, live osteotendinous reflexes, to have anesthesia with T9 sensitivity level, a vibratory disorder in the lower limbs, abolished sphincteric content, and afebrile SaO_2_= 94% in atmospheric air. The patient was discharged with a permeable urinary catheter and home treatment with cortico-therapy and vitamin B1 complex in combination with vitamin B6 and B12.

#### 2.2.2. Case 2

A 78-year-old patient with known paroxysmal atrial fibrillation in treatment, hypertension, and gout was admitted to the neurology department for a crisis of loss of consciousness at home, a language disorder, and a right-limb motor deficit with an unspecified onset. Upon objective neurological examination on admission, the patient was conscious, cooperative, with head and eyeballs deviated to the left, global aphasia, right hemiplegia, and right Babinski.

On admission, the patient had a brain CT scan, which showed acute ischemic stroke in the left MCA with an ASPECTS score of 9 points. On day 2 of hospitalization, the patient underwent a brain MRI, which showed acute ischemic stroke in the superficial and deep left Sylvian territories, in the superficial border territories of the middle cerebral artery/posterior cerebral artery, and in the middle cerebral artery/left anterior cerebral artery, with a small area of hemorrhagic transformation in the lenticular nucleus (Figure 2). On day 7 of hospitalization, the patient developed a cough with mucopurulent sputum, became trachea–bronchial loaded, and was tested by RT-PCR for SARS-CoV-2, with the result being positive. The patient was transferred to a hospital for supporting COVID-19.

#### 2.2.3. Case 3

A 57-year-old patient with a known operated colon neoplasm, essential hypertension, and congestive heart failure was admitted for confusion syndrome and involuntary right-upper-limb movements. Objective neurological examination showed the patient to be conscious, less cooperative, and temporospatially disoriented, with apparently normal oculomotor coordination and no motor deficits; no issues were revealed with coordination and sensitivity tests.

At the time of admission to the emergency care unit, he underwent a native brain CT scan, which showed calcified atheromatous plaques located at the level of the intra- and supracavernous segments of both internal carotid arteries as well as at the level of the intracranial segment of the left vertebral artery and the right maxillary mucoceles, and a deviation of the nasal speculum with an “S” appearance, and leukoaraiosis.

On day 6, the patient underwent a native brain MRI showing multiple infra- and juxta-centimetric lesions in hyper seminal T2/FLAIR, without diffusion restriction, bilaterally arranged in the hemispheric white matter subcortical frontal–temporal–parietal, as well as in the right cerebellar hemisphere and supratentorial demyelinating lesions, most likely with ischemic vascular substrate and linear and curvilinear tracts in a SWAN (susceptibility-weighted angiography)-like manner, and left-frontal and left-parietal cortical hemosiderosis (Figure 3).

On day 10 of hospitalization, the patient underwent EEG (no epileptiform graph elements during recording; involuntary movement in the right-upper limb during investigation without electrical correspondent) and brain MRI angiography with venous time determined to be within the normal limits.

On day 11, he was tested by RT-PCR, having been in contact with a positive COVID-19 case 7 days prior and testing positive himself. During hospitalization, he received hydro-electrolytic rebalancing, and hypotensive, antiepileptic, antibiotic, analgesic, and antiemetic treatment. The patient was transferred to a COVID-19 support hospital after testing positive for SARS-CoV-2 infection.

#### 2.2.4. Case 4

A 69-year-old patient with known type II diabetes, chronic kidney disease grade 3B, hypertension, diabetic nephropathy, and a left-thigh amputation, presented to the emergency care unit for right-limb motor deficit and speech impairment with onset occurring during the previous day.

Neurological objective assessment on admission showed the patient to be conscious, the head and eyeballs to be deviated to the left, with mixed aphasia, right hemiplegia, and predominantly brachial SM = 1/5 and IM = 4/5. Brain CT at the time of presentation showed supratentorial sequelae. On day 6 of hospitalization, the patient underwent a brain CT which showed an unchanged CT appearance compared to the previous examination. On day 8, he had an EEG, which showed a left frontal–temporal–occipital injury with frontal–temporal–left temporal flattening and left temporo–occipital theta waves (Figure 4).

On day 12 of hospitalization, he underwent brain MRI scanning, which revealed left frontal–parietal and left parieto–occipital cortico–subcortical porencephalic–gliotic lesions with dilatation of the adjacent interdigital spaces and a slight retractile effect on the left LV and chronic ischemic ramolitic lesions in the left superficial Sylvian territory and in the superficial border territory of the left middle cerebral artery–left posterior cerebral artery, in the left internal carotid artery with absent circulating flow signal in intracranial segments (the remaining large basal brain arteries were without detectable intralumenal signal changes on parenchymal sequences), and left internal carotid artery occlusion. On day 17 of admission, the patient was tested by RT-PCR, which showed a positive result, resulting in the patient being transferred to a COVID-19 support hospital.

#### 2.2.5. Case 5

A 68-year-old female patient, known to have stroke sequelae and Alzheimer’s disease in the dementia stage, was brought in for subtended tonic–clonic seizures. Neurological assessment on admission showed the patient to be conscious, uncooperative, to have reactive intermediate pupils, present bilateral corneal reflex, left-limb spasticity, apparent upper and lower limbs falling equally to the bed plane, showing right hemicorporeal motor Jacksonisms during consultation, and possessing sacral region decubitus scars. At the time of presentation in the emergency department, the patient presented a positive result for SARS-CoV-2 RNA as tested by RT-PCR.

Chest CT angiography and PET were carried out for the pulmonary arteries and the native brain CT showed cerebellar abiotrophy. On day 5 of hospitalization, the brain CT was repeated, which showed an unchanged appearance from the previous CT examination (cerebellar abiotrophy; in observation, corpus callosum dysgenesis). On day 10 of hospitalization, a brain MRI was performed, which showed an imprecisely delimited area of intense T2/FLAIR signal, slightly restrictive in diffusion, located cortico-subcortically at the level of the left tonsil, late subacute infarction in the territory of the anterior choroidal artery (Figure 5).

Neurological examination at discharge showed the patient to be cachectic, conscious, cooperative, to sporadically execute simple orders and only partially respond to simple questions, with the right hemiparesis 3/5 equally distributed, left hemiparesis spastic sequelae; swallowing was possible, and sacral scars were present. Laboratory tests changed during hospitalization, showing increased D-dimers, thrombocytopenia, increased CK-MB, increased alkaline reserve, increased urea, and hypokalemia. The patient was discharged with home treatment.

#### 2.2.6. Case 6

This patient, aged 66 years, presented to the emergency department for language disorders with fluctuating evolution. Neurological objective examination on admission showed the patient to be conscious, cooperative, with no motor deficits, lower limb ataxia, and bilateral plantar–indifferent cutaneous reflex. During hospitalization, he underwent a diabetes consultation, after which insulin therapy was initiated, a carotid Doppler echo, showing bilateral carotid atheromatosis, an EMG, which showed predominantly sensory axonal polyneuropathy, a cervical–thoracic spine MRI which showed T7–T9–T10 intraspongious herniation and a C5–C6 disc overhang; there was also a C6–C7 protrusion with left C7 radicular conflict, and a brain MRI evidencing T1–T2 nonhomogeneous hyperintense material partially occupying the transverse sinus and left–sided sigmoid sinus, with extension to the jugular bulb–left transverse–sigmoid–jugular venous subacute thrombosis (Figure 6).

On admission, the patient had a negative RT-PCR test, but on day 16 of admission, the patient tested positive for SARS-CoV-2 RNA by RT-PCR, so the patient was transferred to the hospital providing COVID-19 support.

#### 2.2.7. Case 7

A 71-year-old patient, known to have arterial hypertension grade III basal cell epithelioma, presented with two generalized tonic–clonic seizures, without sphincter relaxation, without biting the tongue, for which he was admitted for further investigations and specialist diagnosis. Neurological objective examination on admission showed the patient to be conscious, postcritical, noncooperative, with no neck roll, normal oculomotricity, preserved reflexes, equal intermediate pupils reactive to light stimuli, and no apparent motor deficits.

At the time of presentation, an RT-PCR test for SARS-CoV-2 was positive, a cord–pulmonary X-ray showed bilateral apical sequelae, bilateral interstitial fibrotic changes, normal sized cord and calcifications in the aortic button; a brain CT showed a left-frontal space-replacement formation, and a secondary lesion was observed.

During hospitalization, the patient underwent native and contrast-enhanced abdominal MRI, which revealed a pancreatic cephalic expansive–infiltrative formation with gastroduodenal infiltration and locoregional adenopathy and a simple LDH cyst, and a contrast brain MRI showed a cystic solid mass with peripheral annular gadophilia, axial dimensions of 23/20 mm, extensive perilesional oedema, and a left midfrontal subcortical–left-frontal expansive lesion, most likely of secondary determination significance (Figure 7).

Biological results revealed altered tumor markers (CA 19-9 = 78.9, CEA = 9.9, NSE = 40.1), corrected hypokalemia, corrected hyponatremia, altered urinalysis, elevated cardiac enzymes, biological inflammatory syndrome, and elevated blood glucose. During hospitalization, he received treatment with cerebral antiedematous, corticosteroid, antiplatelet, statin, gastric protector, analgesic, antiemetic, antiepileptic, myorelaxant, hypotensive, diuretic, beta blocker, and antiviral. Neurological objective assessment at discharge showed the patient to be conscious, partially temporospatially oriented, and with no motor deficit. He was discharged with antiplatelet, statin, and antiepileptic medications to be taken at home.

#### 2.2.8. Case 8

A 79-year-old female patient, known to have hypertension and insulin-dependent type II diabetes mellitus, presented to the emergency care unit initially with hypoglycemia (blood glucose = 50 mg/dL) and subsequently presented with a right-limb motor deficit, which was predominantly brachial. Neurological objective examination on admission showed the patient to be conscious, cooperative, partially temporospatially oriented, with symmetrical facies, and predominant brachial right hemiparesis. At the time of presentation, she had a brain CT scan showing a CT appearance within age limits, a cord–pulmonary X-ray showing no evolving pleuropulmonary lesions, and a negative RT-PCR test.

During hospitalization, she underwent a carotid Doppler echo, which evidenced bilateral carotid atheromatosis, and a brain MRI, which evidenced an intraventricular FLAIR intense signal beach, moderately restrictive in diffusion, located in the cortical–subcortical left superior occipital subacute infarction (Figure 8). Chest CT was performed to monitor the lung lesions and specific inflammatory samples were collected on the recommendation of the infection specialist, with whom repeated consultations were carried out to establish therapeutic management.

A chest CT showed alveolo-interstitial infiltrates with a tendency to confluence in bilateral lungs, compatible with COVID-19 pneumonia image-wise, CO-RADS 6-score severity (4/25p), slight pulmonary micronodules with sequelae appearance, and vesicular lithiasis. Seven days later, the repetition of chest CT showed pleurisy in the small right pleura, bilateral dorso-basal pulmonary condensation processes compatible with atelectatic processes, alveolo-interstitial infiltrates with a tendency to consolidation compatible with COVID-19 pneumonia, CO-RADS 6 image-frameable dimensional progression from a previous CT scan, a moderate severity score (11/25p), bilateral pulmonary micronodules with sequelae, and gallbladder lithiasis. An RT-PCR test was conducted, and the result was positive. A biological assessment revealed mild corrected hypokalemia, UTI with treated proteus mirabilis, hyperglycemia, and cardiac enzymes slightly altered.

During hospitalization, the patient received treatment with cerebral depletive, antiplatelet, then double antiplatelet, statin, hypotensive, diuretic, beta-blocker, antibiotic therapy, EU biotic, insulin, antialgesic, gastric protector, hydration solutions, antiviral treatment, corticotherapy, anticoagulant, initially HGMM, then NOAC, under which the evolution of the patient was favorable. Neurological objective examination at extenuation showed the patient to be conscious, cooperative, partially temporo-spatially oriented, with an MMSE test score = 20 points, no neck rolling, normal oculomotricity, no motor deficits, no coordination or sensitivity disorders, afebrile, no cough, and no dyspnea.

#### 2.2.9. Case 9

A 41-year-old patient, who was a smoker and recently discharged from a medical clinic with a diagnosis of unknown substance intoxication and metabolic acidosis, presented to the emergency care unit due to seizures. The patient also presented himself 3 days prior with headaches, whereby he experienced a seizure during his admission.

Neurological objective examination on admission showed the patient to be conscious, cooperative, with no neck rolling, biting tongue, nystagmus was exhaustible on right lateral gaze, normal oculomotor coordination, left hemiparetic whip 4/5, osteotendinous reflexes were present, symmetrical, ataxia on left heel–knee test. During hospitalization, native and contrast-enhanced brain CTs were performed, showing right area, parietal encephalomalacia area, and old consolidated bilateral parietal fractures. Brain MRI showed an oval mass with dimensions of about 45/24/46 mm, isointense fluid content with cerebrospinal fluid, normal water diffusion, and negative gadophilia, located subcortically in the right parietal; the lesion was circumscribed and crossed by several very fine vascular tracts, did not communicate with the ventricles or the subarachnoid space, associated with the perilesional plaques of the intense FLAIR signal, and exerted a slight mass effect on the right lateral ventricle in the right parietal cystic expansive process (Figure 9).

On day 10 of hospitalization, the RT-PCR test was positive. Biological changes showed discrete mixed hyperlipidemia. During hospitalization, antiepileptic and antiedema cerebral treatment was administered. Neurological objective examination at discharge showed the patient to be conscious, cooperative, and temporospatially oriented, with no neck rolling, normal oculomotor function, no motor deficit, and no sensitivity or coordination disorders, and walking without support was possible.

#### 2.2.10. Case 10

A recently infected 72-year-old elderly patient with a severe form of COVID-19 was transferred from the Cardiology department with acute fibro flutter, and a recent brain CT (admission from Infectious Diseases Hospital) showed left ischemic PCA stroke appearance, with a brain MRI describing a left ischemic ACP stroke that transformed into hemorrhagic stroke, present during acute COVID-19 infection and post-hospitalization at home with bilateral lower limb plegic motor deficit.

Neurological objective assessment on admission showed the patient to be conscious, partially cooperative, and partially temporospatially oriented, with a right-upper-limb motor deficit, flaccid paraplegia, abolished osteotendinous reflexes in the lower limbs, and no tactile surface sensitivity disorders. During hospitalization, she underwent native and contrast-enhanced chest CT scanning, which showed alveolo-interstitial changes suggestive of post-COVID-19 status, lumbar puncture (CSF macro- and microscopically normal appearance), and a thoracolumbar spine MRI showing a left minor split disc L5-S1 with a subligamentary nonmigrated pulposus nuclear fragment. Biological tests revealed liver cytolysis, hypokalemia, and corrected hyperglycemia. During hospitalization, she was treated with statin, antiplatelet, and hydration infusion solutions, with unfavorable evolution. The patient suffered cardio-respiratory arrest (Figure 10, Figure 11, Figure 12, Figure 13 and Figure 14).

### 2.3. Neurological Features of Patients

Table 2 describes the baseline and clinical characteristics of COVID-19 patients with neurological features.

## 3. Discussion

The COVID-19 patients in our 10 case reports presented with a complex panel of neurological diagnostics, including myelitis with paraparesis, acute ischemic stroke in various territories with hemiplegia, cerebrovascular disease with involuntary movements, seizures, and left transverse–sigmoid–jugular venous subacute thrombosis.

Cerebrovascular disease, in addition to certain other neurological features, has often been associated with acute SARS-CoV-2 infection. Numerous pathophysiological mechanisms have been postulated to explain the SARS-CoV-2-related prothrombotic condition, as both direct and indirect consequences of the viral infection. Aside from hypercoagulable characteristics, it is hypothesized that SARS-CoV-2-related endothelitis and microangiopathy lead to hemorrhagic stroke. Consequently, intracranial hemorrhage in COVID-19 patients could be the result of hemorrhagic transformation of ischemic stroke, original hemorrhagic stroke, or traumatic intracranial hemorrhage.

The processes underlying the apparition of these neurological symptoms remain unknown. Numerous ideas have been advanced since SARS-CoV-2 was first detected, such as that the neuroinvasion of the virus comes from its ability to enter via the olfactory groove or directly into the nervous system via circulation [6,7]. However, these results might be the result of secondary immunological processes and a severe inflammatory state induced by infection, or of significant hypoxia caused by critical illness and concomitant disorders [6,7].

We have identified various neurological pathologies which can be correlated to the positive diagnosis of the RNA SARS-CoV-2 testing by RT-PCR tests or which represent only incidental findings following a clinical or paraclinical examination by magnetic resonance imaging.

The types of disorders encountered during the study are acute stroke, subacute stroke, hemorrhagic stroke, ischemic stroke that then becomes hemorrhagic, carotid transitory ischemic stroke, vertebrobasilar insufficiency, cerebrovascular disease, venous thrombosis, demyelinating lesions, sequela lesions, secondary determinations, tumoral formation, myelitis, seizures, Guillain–Barre syndrome, paraesthetica syndrome, paraparesis, myasthenia gravis, multiple sclerosis, Rasmussen’s encephalitis, movement lacunar stroke, amnestic syndrome, and disk protrusion.

The most frequent pathology was ischemia, which is strongly connected to a diagnosis of COVID-19 and is represented by cerebral strokes with various ages and types of evolution, with a total number of 85 cases of cerebral strokes, 44 cases of acute cerebral strokes, 36 cases of subacute cerebral strokes, and 5 cases of carotid transitory ischemic stroke. This classification might also include four cases of ischemic ictus, two cases of movement lacunar stroke, two cases of amnestic syndrome, three cases of stroke sequela, and two cases of sequela lesions. Among the patients who had been diagnosed only with the cerebrovascular disorder, we found six cases and four cases of demyelinating lesions, which might have had a vascular-ischemic sublayer. We also encountered a case of vertebrobasilar syndrome.

The hemorrhagic stroke associated with the SARS-CoV-2 virus occupied a significant place in a group of 26 cases, of which 20 patients presented with a hemorrhagic acute stroke and another 6 patients presented with a hemorrhagic stroke transformed from an ischemic stroke.

Moreover, the neoplasia pathology associated with COVID-19 had substantial representation, as we found three cases with tumoral formations and five cases with secondary determinations. We found two cases of venous cerebral thrombosis associated with SARS-CoV-2 infection and three cases of convulsive crisis.

The range of neurological pathologies associated with the infection of SARS-CoV-2 is vast, and we might include other pathologies here, such as myelitis, Guillain–Barre syndrome, paresthesia, paraparesis, myasthenia gravis, multiple sclerosis, Rasmussen’s encephalitis, and disk protrusion.

Ghannam et al. indicated that infection with the SARS-CoV-2 virus results not only in respiratory disease, and that neurological complications are frequently experienced. Associated neurological pathologies are often encountered, including ischemic and hemorrhagic strokes, Guillain–Barre syndrome, encephalitis, and convulsions. In a studio located in Wuhan, China, out of 214 cases with SARS-CoV-2 infection, 78 had neurological complications. Patients with severe SARS-CoV-2 infection have a higher risk of developing neurological complications. The manifestations point to the central nervous system and peripheral nervous system, and symptoms such as headaches, vertigo, muscular weakness, sensory alteration, and impaired consciousness may also appear [8].

As part of the classification according to age groups, it was noticed that the greatest number of cases were of patients aged 70–80 years, representing 58 out of the 150 cases that were evaluated.

This age group was followed by 60–70-year-olds, with 40 cases, followed by the 80–90-year-old age group, in which we found 26 cases. In these first three groups, ischemic vascular pathology was most frequently presented. Next, we found the group of 50–60-year-olds, with 17 cases, where inflammatory pathologies and venous thrombosis were the most frequently presented.

For the other age intervals, the number of cases decreased dramatically, with one case for the group of 30–40 years, seven cases for 40–50 years, and one case for 90–100 years.

Sullivan et al. evaluated neurological pathologies according to patient age, individual age, or the age interval of the group, with the aim of classifying the frequency of neurological complications associated with SARS-CoV-2 and that affect children (under the age of 19 years), young adults (between 19 and 50 years), and adults (over the age of 50). The patients with an average age between 60 and 69 years presented the greatest number of associated neurological disorders, and the patients aged equal to or under 9 years presented the lowest association. Patients over the age of 50 years presented the greatest number of cerebrovascular disorders. The most frequent associated neurological pathology was observed in over half of the patients in this group [9].

Patients with COVID-19 displayed a predominance of respiratory and nervous system symptoms. Among several patients of the latter, there were complaints of loss of smell and taste, ataxia, and impairment to peripheral nervous system, which might reflect the SARS-CoV-2 virus presenting with neurotoxicity.

Six key clock genes, including *CLOCK*, *BMAL1*, *PER1*, *PER2*, *CRY1**,* and *CRY2*, control circadian rhythm. Clock genes are involved in the regulation of metabolic and immunological responses, including the release of pro-inflammatory interleukins. As a result, lifestyle modifications, including adjustments in the light regime, a reduction in the amplitude of room temperature, a change in the timing of eating, and the allocation of a food according to its calorific value throughout the day, contribute to metabolic abnormalities and the apparition of a low-intensity systemic inflammatory process [10]. Additionally, the circadian clock governs fundamental bodily processes, including lung capacity and sleeping as well as activities in the neural tissue that contribute to neurological and psychiatric illnesses, such as auto-aggression and neuropathic pain. Additionally, several studies have discovered a link between the development of certain symptoms and particular circadian chronotypes that might aid in the creation of chronotherapy and enhance therapy by administering medication in line with the patient’s circadian rhythm [11,12].

Sleep phase abnormalities were the most prevalent circadian rhythm abnormalities. Boiko et al. [11] discovered that COVID-19 infection in antecedents increased patients’ susceptibility to developing circadian rhythm abnormalities, including delayed sleep phase disorder. Additionally, it was shown that individuals with COVID-19 exhibit elevated levels of trait and state anxiety. The high incidence of delayed sleep phase disorder might be a result of both the neurotropic properties of SARS-CoV-2 and isolation linked with the COVID-19 outbreak. Therefore, the impact of a modification in light exposure on everyday activities was detected under lockdown settings, meaning a change of the rhythm by three or more hours. This has resulted in decreased exposure to natural light, which has a detrimental effect on the main circadian oscillator, as well as physical and nutritional activities. Amongst circadian rhythm abnormalities, delayed sleep phase disturbance is most prevalent in the post-COVID-19 era and is related to an increased degree of anxiety in these individuals.

### Global Impact of the COVID-19 Pandemic on Different Areas in Low- and Middle-Income Countries

Formerly, it was thought that the SARS-CoV-2 virus was limited to the respiratory system, mostly impacting the lungs; nevertheless, new investigations have demonstrated the virus’s multisystem impact, most notably impacting brain tissue. COVID-19 illness may indeed be associated with a variety of mental disorders, including post-traumatic stress disorder, obsessive–compulsive disorder, anxiety, delirium, and depression, either directly or indirectly. It has the potential to exacerbate pre-existing mental health problems or to precipitate the genesis of new psychiatric diseases [13]. Increasing COVID-19 instances, elevated illness burden, and a lack of social support may all contribute to short-term mental health concerns, while economic losses also present because mandated lockdowns could have a long-term effect on people’s mental health [14].

Due to a global scarcity of frontline staff dedicated to containing the epidemic, several governments have reassigned psychiatrists to critical care settings to handle COVID-19 patients. As a result, mental hospital outpatient departments were closed, resulting in a massive increase in mental health difficulties in some countries. In these cases, imposed lockdowns and restrictive restrictions regarding physical isolation have robbed many patients of access to essential mental health treatment [15].

The COVID-19 pandemic has highlighted underdeveloped health systems in low- and middle-income countries (LMICs) and significant treatment disparities, at least in the area of mental health. As a result of these issues, individuals with severe mental illness die earlier, have more physical ailments, and receive less medical treatment than the general population [16].

Additionally, it is vital to educate the public in low- and middle-income countries on the acknowledgment of mental health problems as diseases, the importance of social and familial assistance, and the importance of avoiding social stigma of those who suffer from mental health disorders.

Most neurological symptoms have been shown to develop early throughout the course of the disease (in some studies, the median time to hospital admission was 1–2 days) [15,16]. Several individuals lacking classic COVID-19 manifestations (fever, coughing, anorexia, and diarrhea) presented to the hospital with solely neurological manifestations, as in our study. As a result of the impact of SARS-CoV-2 infection on neurological diseases, we must closely monitor patients with COVID-19 for neurological symptoms, particularly those with serious infections that could have led to their mortality. Additionally, during the COVID-19 epidemic era, clinicians must include SARS-CoV-2 infection as a differential diagnosis whenever meeting patients with these neurological signs to minimize late or incorrect diagnosis and to limit the spread.

The purpose of this research was to provide a complete assessment of neurological symptoms related with SARS-CoV-2 infection and to detail the course of disease and outcomes of COVID-19 individuals who developed neurological symptoms in 10 representative patients from Constanta Clinical County Hospital, which is situated in a low–middle-income country. This work may provide critical new clinical information on COVID-19, assisting physicians in raising awareness of its association with neurological symptoms and diagnosis of COVID-19 infection in low- and middle-income countries.

It is particularly significant to note that, in our study, contrary to other studies, the patients firstly presented with neurological features, and COVID-19 infection was subsequently discovered. Moreover, quick clinical decline or aggravation may be accompanied with a neurological event, including stroke in patients with severe COVID-19, contributing to the disease’s high mortality rate. Additionally, doctors might include SARS-CoV-2 infection as a differential diagnosis when meeting patients with these neurological signs during the COVID-19 pandemic period to prevent late diagnosis or misdiagnosis as well as to limit the spread. To facilitate future clinical care, additional precise epidemiological data and further pathophysiology findings are required.

Global restrictions imposed to prevent and control the spread of a new COVID-19 wave resulted in a financial crisis in LMICs, restricting access to food and other basic requirements due to border closures. The combined consequences of poverty, climate change, and the COVID-19 epidemic have exacerbated food insecurity in some LMICs. COVID-19, thus, worsened an already-existing food crisis in these countries due to the imposition of government restrictions and lockdown measures that restricted work options and income.

Food insecurity is a major socioeconomic and public health problem in low- and middle-income nations. It is associated with adverse health effects and a reduction in self-reported health status, decreased micronutrient intake, fruit and vegetable consumption, weight gain, and birth abnormalities. As with challenges experienced by other resource-related issues (e.g., housing instability, energy uncertainty), inequality and poverty might exacerbate nutritional deficiencies, illness, and disease management. Individuals who experience poverty have much worse health outcomes and less access to healthcare than those who do not. Poor nutrition may exacerbate pre-existing illnesses, such as inadequate glucose control in diabetic patients, final renal disease in patients with chronic kidney disease, as well as affect the treatment of other chronic diseases. COVID-19′s presence in LMICs has restricted access to healthcare and impacted attempts to treat, diagnose, immunize, and monitor other infectious diseases. Food instability may further aggravate health problems and expenditures for families with children that have specific healthcare requirements, or for persons with disabilities.

A number of examples can be provided depicting the impact of the COVID-19 pandemic on LMICs.

Afghanistan’s continuous struggle has created several difficulties for the country’s population. Afghanistan has seen a significant rise in food shortfalls because of its reliance on neighboring nations during the epidemic [17]. Individuals attain food security when they have continuous physical and economic access to an adequate supply of safe and nutritious food that meets their dietary demands and preferences. Food scarcity, political unrest, and the third wave of COVID-19 have made it impossible to obtain basic supplies. Consequently, folks are forced to contend with the COVID-19 pandemic amid economic collapse and despair. At this crucial point, worldwide efforts are essential to ameliorate food security.

The growth in the number of instances of the illness has the potential to overwhelm the health system, as does noncompliance with social distance measures and the introduction of variants of concern in LMICs. This rise in the transmission curve may also create conditions favorable for the emergence of further changes in the virus’ structure and DNA. As a result, genomic monitoring methods are essential to detect and describe these variants as well as to determine if the vaccines against the virus that are currently in use are efficacious.

The development of efficient and dependable infectious disease monitoring systems is critical for establishing a high-quality public healthcare system and reducing the mortality rates in low- and middle-income countries. Monitoring helps facilitate the accessibility of records and knowledge and decreases the burden and propagation of unfavorable healthcare events. This allows for a rapid response in public health, efficient implementation of methods and countermeasures, and a review of suggested treatments, the rapid detection of new illnesses, promoting health security and stability for people living in LMICs.

Healthcare workers (HCWs) have been critical in containing the pandemic and mitigating its effects. Increased working hours and frequent exposure to critically ill patients have major consequences for the health and wellbeing of physicians, which have previously been disregarded.

Infectious disease epidemics have always posed a threat to public health, especially in Africa, where outbreaks have exploded in recent years [18]. Although several infectious diseases have emerged in Africa, such as Ebola and certain other epidemic-prone infections, insufficient focus has been placed on the development of health surveillance systems. The inadequacy of the region’s healthcare monitoring techniques have only recently been identified. Africa suffers from a shifting epidemiology of disease, a deficient healthcare system, and a scarcity of resources. Only a vigilant monitoring system can ensure that the best use of its available resources are made in an effective and strategically managed manner. Measures are needed to rapidly detect potential public health threats. This could be accomplished via the use of appropriate, efficient, and lengthy surveillance methods.

Dengue fever is a serious public health concern in Africa [18] and the COVID-19 pandemic has exacerbated this concern. COVID-19 accelerates the transmission of a variety of illnesses, including Zika, yellow fever, measles, mucormycosis, Lassa fever, and HIV, as has been seen in several nations on the African continent. The limits imposed in response to the COVID-19 pandemic have resulted in the suspension of vector management initiatives that aid in the management of these diseases. To avert further public health disasters, urgent and interdisciplinary measures to dengue fever epidemic management in African nations are necessary.

COVID-19 cases have impoverished Nigeria’s healthcare system and resulted in additional neglect of persons suffering with mental illness [18]. In general, there is a demand for equitable access to healthcare resources, but there is a need for adequate attention and treatment for mental health patients, which is rising in Nigeria.

The COVID-19 epidemic has hit public health emergencies in Bangladesh, a low–middle-income nation in South Asia. The resulting surge of sickness cases may generate an overburdening of the health system. This rise in the contagion curve may also encourage further alterations in the virus’ structure and DNA. It is important to find, monitor, and characterize these polymorphisms and determine the efficacy of existing vaccinations against these variants.

India’s healthcare sector is suffering significant difficulties as a consequence of the lack of resources to combat the COVID-19 pandemic, with HCWs also suffering the consequences [19]. There is an urgent need to address these flaws in the healthcare system to provide a consistent and ongoing supply of high-quality treatments in India.

## 4. Conclusions

The neurological manifestations associated with SARS-CoV-2 virus infection are frequent, with ischemic stroke being the most common, followed by hemorrhagic stroke and neoplasia pathology. In this study, we included primary tumoral formations or secondary determinations. In a small number of cases, other disorders were identified, such as myelitis, convulsive crisis, Guillain–Barre syndrome, paresthesia, paraparesis, myasthenia gravis, multiple sclerosis, and Rasmussen’s encephalitis. These could be related to the inflammatory response or even the hypercoagulability caused by the virus, or simply to the virus’s cytopathic effect.

Another conclusion we can draw from the study is that the most affected age group was found to be between 70 and 80 years, representing 58 of the total 150 cases studied, followed by the 60–70 age group, with a total number of 40 cases, followed by the 80–90 year-old age group, with 26 cases.

COVID-19 patients often have neurological symptoms. Throughout the duration of the COVID-19 epidemic, health professionals must consider acute respiratory syndrome SARS-CoV-2 infection as a differential diagnosis in the cases of patients with neurological symptoms to minimize postponed or incorrect diagnosis and loss of the opportunity to prevent and control additional transmission.

The study is limited and observational. More studies aimed at clarifying the neurological complications associated with the infection with the SARS-CoV-2 virus are needed to establish the physiopathological mechanisms.

## Figures and Tables

**Figure 1 diagnostics-12-00473-f001:**
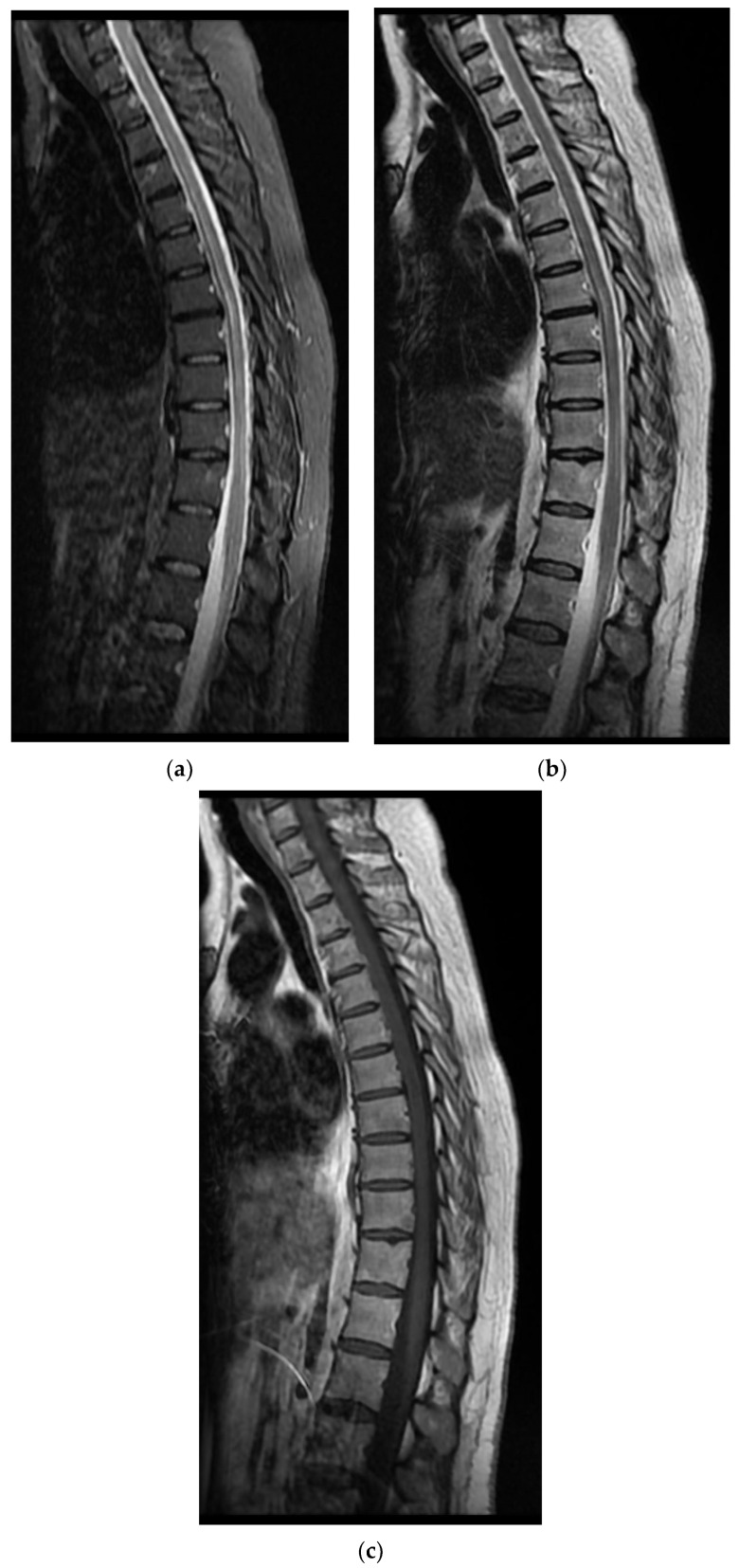
MRI images of the Case 1 patient show inaccurate delimited discrete hyperintense foreshores T2-STIR (**a**,**b**), hypointense T1 (**c**), disposed at the level of lateral coordinates of the marrow corresponding to T6–T10 myeloma; thoracic intramedullary lesions with suggestive characters for an inflammatory-infectious sublayer.

**Figure 2 diagnostics-12-00473-f002:**
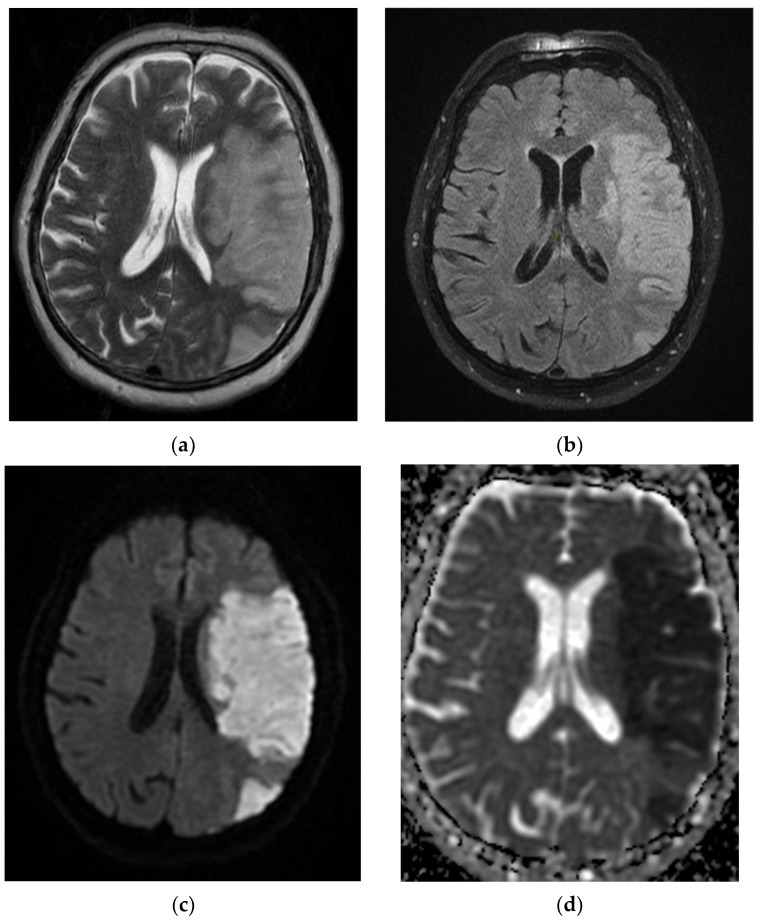
MRI images of the Case 2 patient shows extended foreshores, irregularly delimited, T2-FLAIR hyperintense (**a**,**b**), with restriction of marked diffusion (**c**,**d**), confirmed by a hypointense signal on the map of apparent diffusion (**d**), situated in the cortical–subcortical frontal–parietal–occipital and left insular part, with capsular and lenticular extension. Centimetric intralesional focal signal point SWAN sequence was associated in the left lenticular nucleus (**e**). These characteristics are suggestive of acute ischemic stroke on the left superficial and profound Sylvian territory, and on the superficial territories of the border between the median cerebral artery/posterior cerebral artery and medium cerebral artery/left anterior cerebral artery (**a**–**d**), with a small area of hemorrhagic transformation in the lenticular nucleus (**e**).

**Figure 3 diagnostics-12-00473-f003:**
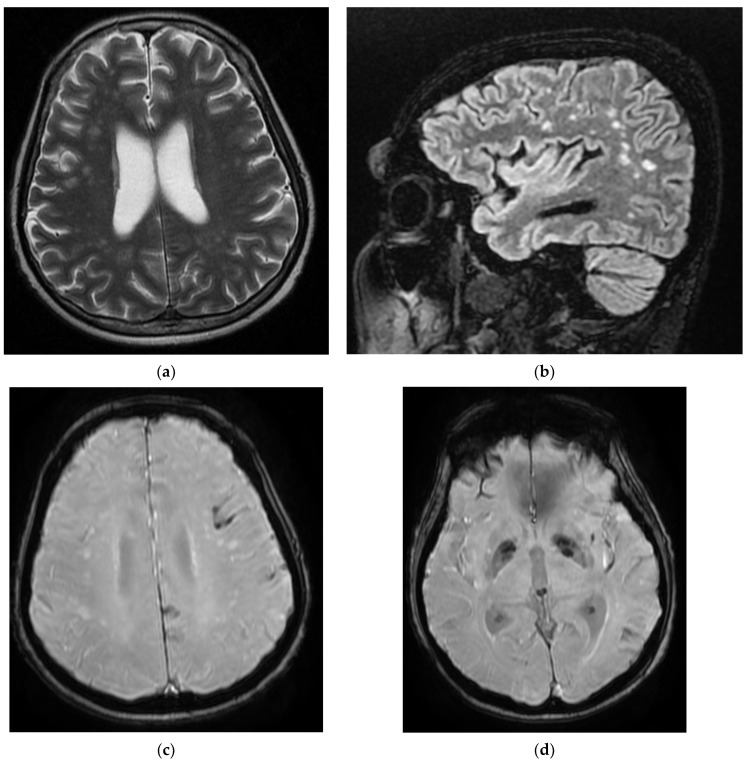
MRI images of the Case 3 patient reveal multiple centimetric lesions in T2/FLAIR hypersignal (**a**,**b**), with no diffusion restriction, disposed in a white hemispheric substance in the bilateral subcortical frontal–temporal–parietal area, as well as in the right cerebellar hemisphere, and supratentorial demyelinating lesions most probably with an ischemic vascular sublayer. Additionally, linear and curvilinear traces in SWAN signal (**c**,**d**), disposed in the left cortical parietal, frontal area–frontal cortical and left parietal hemosiderosis.

**Figure 4 diagnostics-12-00473-f004:**
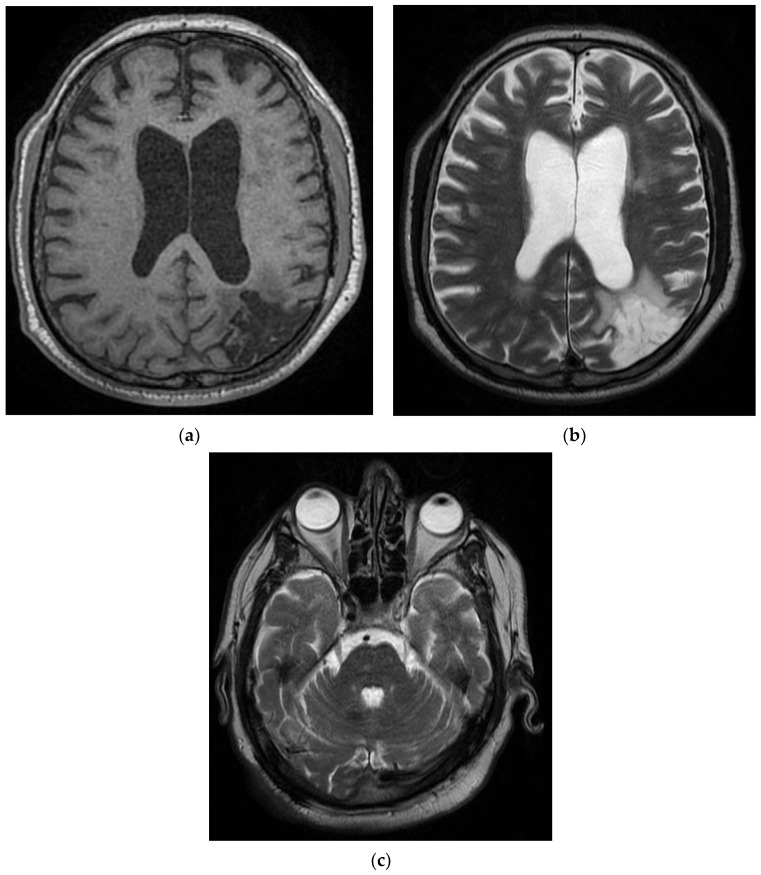
T1 and T2 MRI sequences in the Case 4 patient reveal left porencephalic–gliotic cortical–subcortical parietal–occipital, with a minor retractile effect on the left lateral ventricle; ischemic chronic softening on the left Sylvian territory and on the superficial territory at the border of the median cerebral artery–left posterior cerebral artery (**a**,**b**). The T2 MRI sequence shows the left internal carotid artery without a signal of circulatory flow in the intracranial segments (**c**); the rest of the big arteries located at the bases of the brain with no modifications of intraluminal signal that can be observed on the parenchymal sequences; occlusion of the left internal carotid artery (**c**).

**Figure 5 diagnostics-12-00473-f005:**
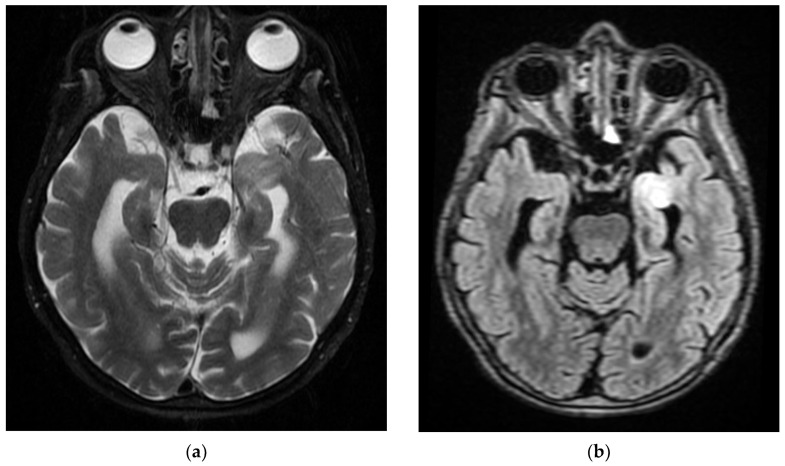
Cerebral MRI examination of the Case 5 patient highlights inaccurately delimited area of intense T2 and FLAIR signal (**a**,**b**), slightly restrictive in diffusion (**c**,**d**), situated in the cortical and subcortical area of the left tonsil. These images are conclusive for subacute tardive infarction on the territory of the anterior choroidal artery.

**Figure 6 diagnostics-12-00473-f006:**
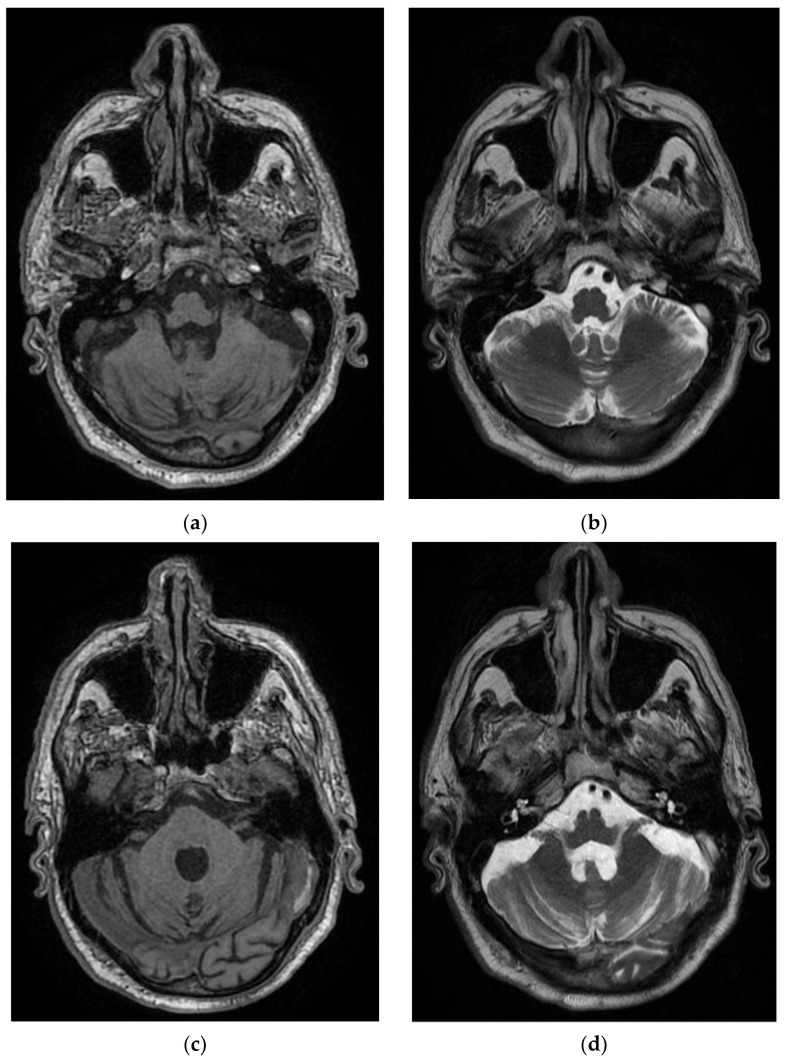
Hyperintense and nonhomogeneous T1 and T2 material, in the Case 6 patient, which partially occupies the transverse sinus (**a**,**b**) and left sigmoid sinus (**c**,**d**) towards the jugular bulb (**e**,**f**)–left venous transverse–sigmoid–jugular subacute thrombosis.

**Figure 7 diagnostics-12-00473-f007:**
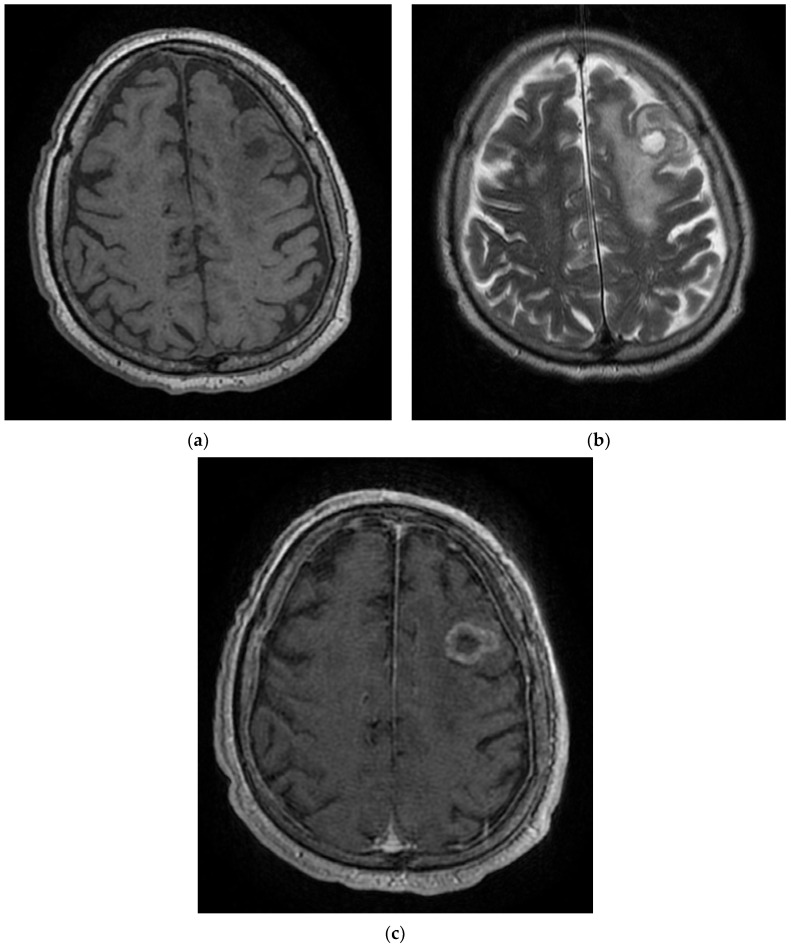
MRI T1 (**a**), T2 (**b**), and T1 sequence with contrast injection in the Case 7 patient (**c**) show solid-cyst mass (**a**,**b**) with ring-shaped peripheral gadophilia (**c**) and extended perilesional edema (**b**), disposed in the middle-left subcortical area. These pathological images are suggestive of a left-frontal expansive lesion, most likely meaning a secondary determination.

**Figure 8 diagnostics-12-00473-f008:**
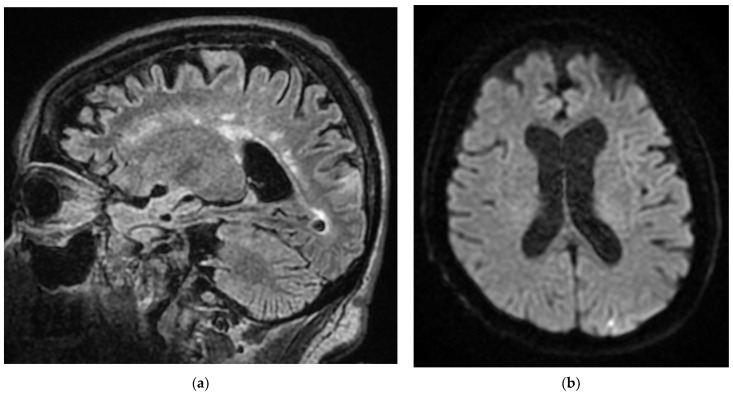
MRI sequences for the Case 8 patient indicate infra-centimetric foreshore of intense FLAIR signal (**a**), moderately restrictive in diffusion (**b**,**c**), situated on the left superior occipital lobe revealing subacute left superior occipital infarction.

**Figure 9 diagnostics-12-00473-f009:**
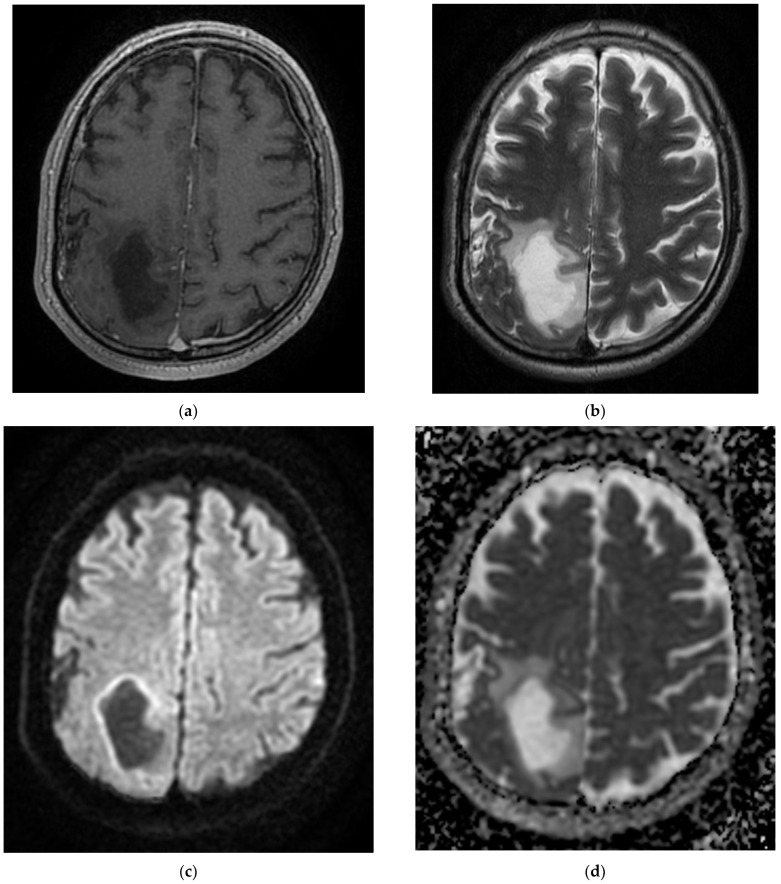
MRI images of the Case 9 patient show oval mass with liquid content (**a**,**b**) and normal diffusion of water (**c**,**d**), located in the right parietal subcortical area; the lesion described is circumscribed and crossed by very thin vascular lines (**e**), and does not communicate with the ventricles nor with the subarachnoid space, associates with the perilesional foreshores of the intense FLAIR signal (**f**), and has a slight mass effect on the right lateral ventricle (**e**). In conclusion, the findings revealed a right parietal cystic expansive process.

**Figure 10 diagnostics-12-00473-f010:**
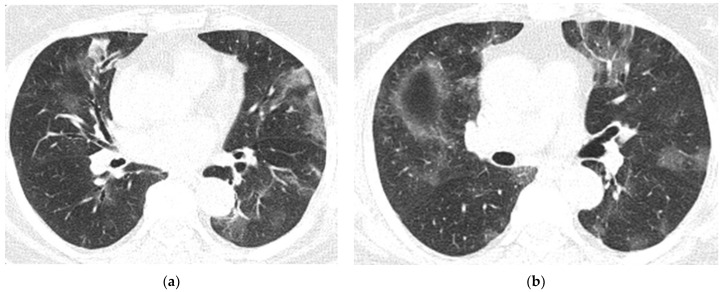
Pulmonary CT images of the Case 10 patient show lesions with a polymorphic aspect, some under the form of inaccurately delimited areas of matt-glass clouding (**a**,**b**), and others with increased densities and associated interlobular septal thickness (**a**), randomly disposed at the level of both pulmonary fields. The tendency was to condensate some lesions from the level of the right anterior basal segment (**a**). Linear fibrosis outlines with retractile effect on the parietal pleura and on some subsegmental bronchial outlines, which could be highlighted especially at the inferior lingual level and at the level of the basal pyramid, bilaterally (**a**,**b**). In conclusion, fiber–alveolar–interstitial modifications are compatible with lesions of SARS-CoV-2 type in various phases of evolution, with a severity score of 13 (8 right lung, 5 left lung), determinable in moderate impairment.

**Figure 11 diagnostics-12-00473-f011:**
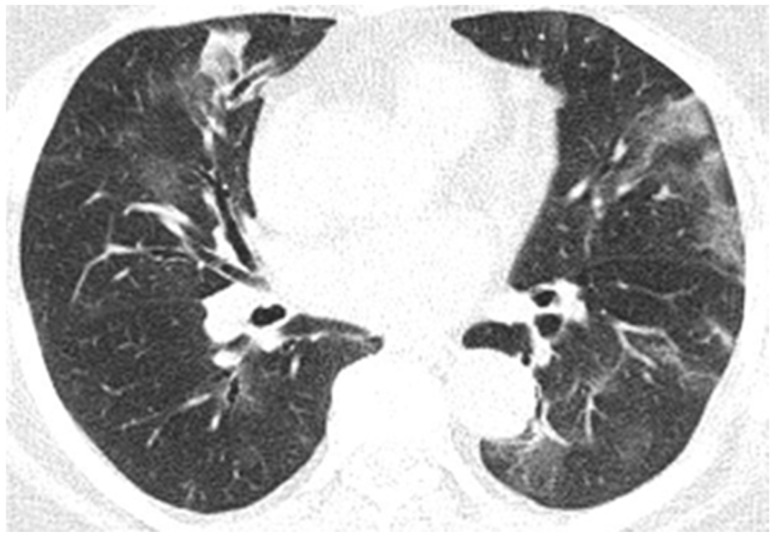
In evolution, CT images of the Case 10 patient show important numerical and dimensional progression of pulmonary lesions randomly distributed on more than 70% of the entire surface of both pulmonary fields. In conclusion, bilateral pulmonary condensations of SARS-CoV-2 type were in progress, with a severity score of 20 (13 at previous examination) and were determined to exhibit a severe degree of disorder.

**Figure 12 diagnostics-12-00473-f012:**
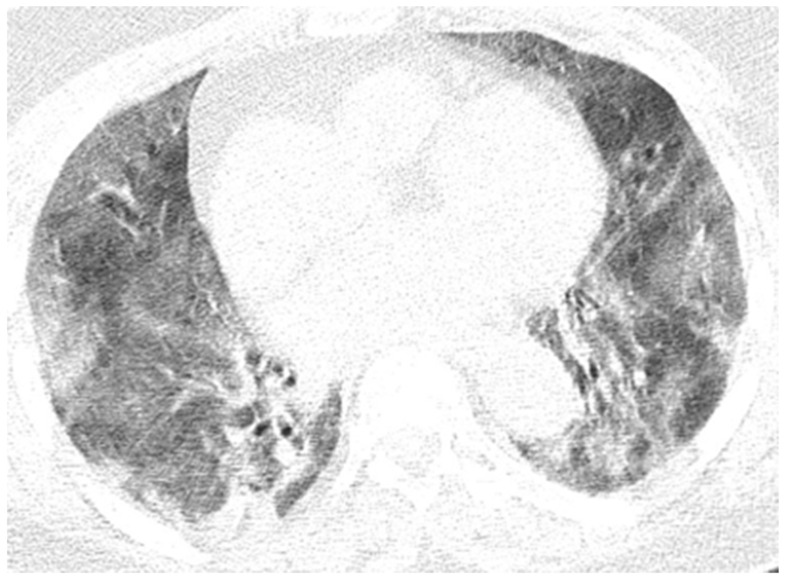
Four days later, pulmonary CT in the Case 10 patient showed dimensional extension of the previously described lesions, with a tendency of small-holding. In conclusion, pulmonary lesions of SARS-CoV-2 type (with the tendency of consolidation) in dimensional progression, with a severity score = 22 (20 for the previous examination), which corresponds to a severe disorder.

**Figure 13 diagnostics-12-00473-f013:**
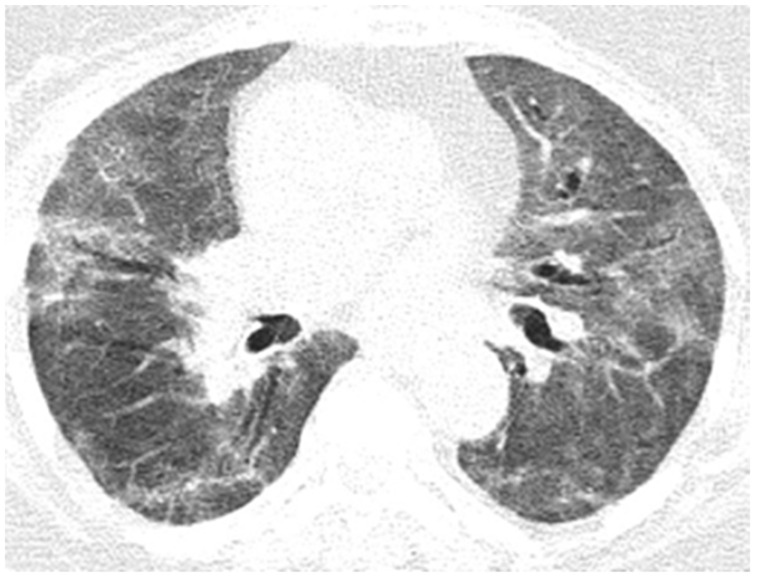
After 10 days, the thoraco-pulmonary CT of the Case 10 patient revealed that the previously described lesions are numerically and dimensionally stationary, at times reduced in intensity. In conclusion, pulmonary lesions of SARS-CoV-2 type in discrete remission, and were severely impaired.

**Figure 14 diagnostics-12-00473-f014:**
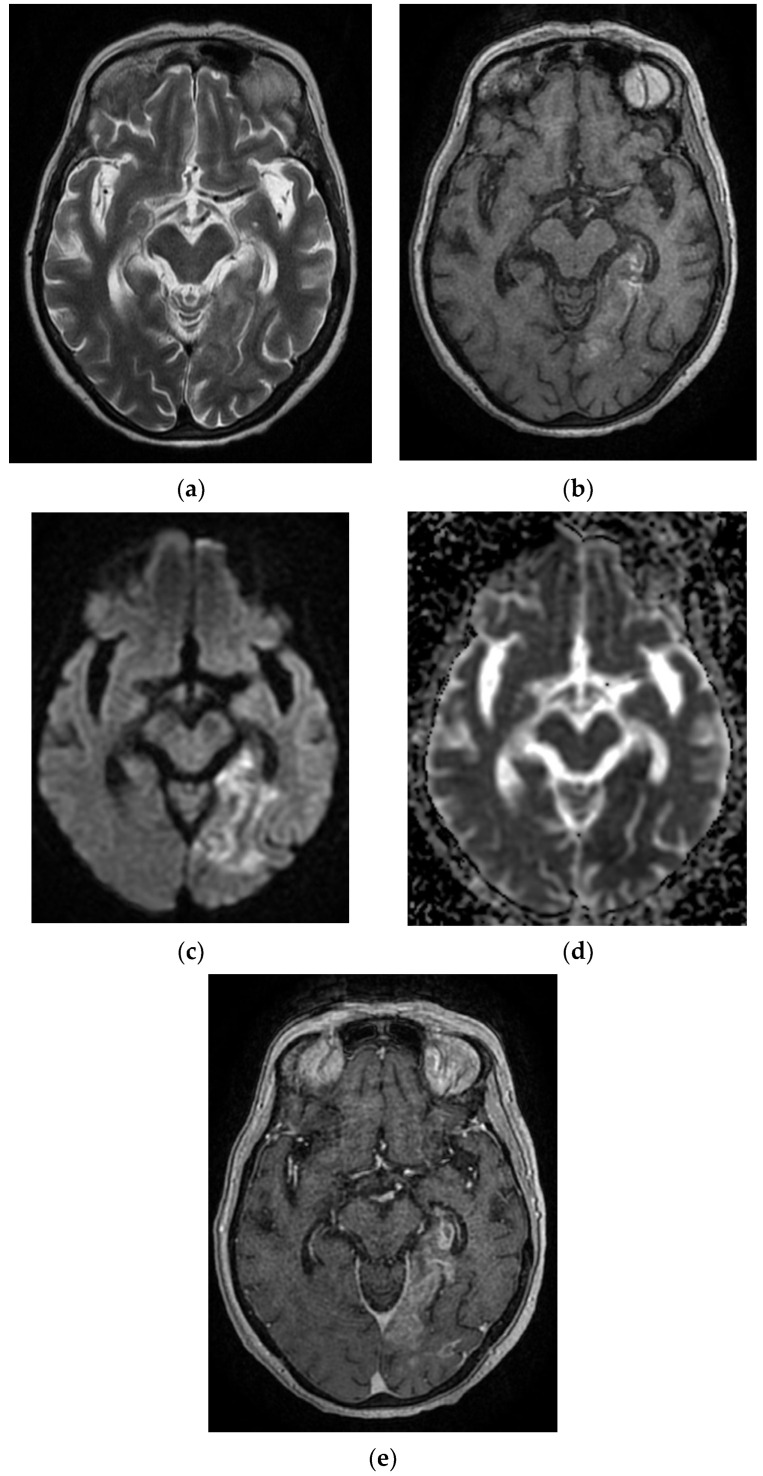
After 11 days from the first pulmonary CT, the cranio-cerebral MRI in the Case 10 patient showed foreshadowing of intense T2 signal (**a**), with deposits of predominantly peripheral methemoglobin (**b**), deficient restriction of diffusion (**c**,**d**) and gyriform gadolinium I (**e**), developed in the cortical–subcortical–occipital left median side with extension in the thalamic nucleus on the same part. In conclusion, infarction hemorrhagically transformed into superficial and profound territories of the left posterior cerebral artery.

**Table 1 diagnostics-12-00473-t001:** Type of neurological disease presented by patients, number of cases, and patient age.

Type of Neurological Disease	Number of Cases	Males	Females	30–50 Years	50–70 Years	70–100 Years
Acute ischemic stroke	44	20	24	-	14	33
Subacute ischemic stroke	36	20	16	-	14	19
Acute hemorrhagic stroke	20	9	11	-	9	11
Hemorrhagic transformation after ischemic stroke	6	3	3	-	2	4
Transient ischemic attack	5	2	3	-	2	3
Vertebrobasilar syndrome	1	-	1	-	-	1
Cerebrovascular disease	6	2	4	-	3	3
Venous thrombosis	2	1	1	1	1	-
Demyelinating lesions	4	2	2	-	1	3
Sequelae lesions	3	2	1	-	-	3
Secondary determinations	5	4	1	-	2	3
Tumor formation	3	2	1	1	1	1
Myelitis	1	-	1	1	-	-
Convulsive seizures	3	2	1	-	1	2
Guillain–Barre syndrome	1	-	1	-	1	-
Paresthesia syndrome	1	-	1	1	-	-
Paraparesis	1	-	1	-	1	-
Myasthenia gravis	1	-	1	1	-	-
Multiple sclerosis	1	1	-	1	-	-
Rasmussen’s encephalitis	1	1	-	1	-	-
Motor lacunar stroke	2	1	1	-	-	2
Amnestic stroke	2	1	1	1	-	1
Disc protrusion	1	1	-	-	1	-

**Table 2 diagnostics-12-00473-t002:** Baseline and clinical characteristics of COVID-19 patients with neurological features.

Case ID	Age	Neurological Features			Other Diagnostics
		Neurological-COVID-19-related diagnostic	Other neurological diagnoses	Neurological symptoms	
1	49	Myelitis; paraparesis		Ascending lower limb paresthesia and lower limb motor deficit	Biological inflammatory syndrome; acute urinary retention; confirmed infection with SARS-CoV-2.
2	78	Acute ischemic stroke in the superficial and deep-left Sylvian territory and in the superficial border territories, middle cerebral artery/posterior cerebral artery, and middle cerebral artery/left anterior cerebral artery by cardioembolic mechanism; right hemiplegia		A crisis of loss of consciousness, language disorder, and right limb motor deficit	Permanent atrial fibrillation; hypertension; gout; biological inflammatory syndrome; interstitial pneumonia; confirmed infection with SARS-CoV-2.
3	57	Cerebrovascular disease; involuntary right upper limb movements	Cerebral atherosclerosis	Confusion syndrome and involuntary right upper limb movements	Dilated cardiomyopathy with left ventricular dysfunction; congestive heart failure class III (New York Heart Association (NYHA)); mild mitral regurgitation; acute chronic kidney disease; upper rectal neoplasm operated and radio chemically treated; hypertension; confirmed infection with SARS-CoV-2.
4	69	Cerebrovascular disease	Sequelae of stroke; right sequelae hemiparesis	Right limb motor deficit and speech impairment with onset	Left internal carotid artery occlusion; hypertension; type II diabetes mellitus with diabetic nephropathy. Chronic kidney disease grade 3B; left-thigh amputation; confirmed infection with SARS-CoV-2.
5	68	Subacute ischemic stroke in the territory of the anterior choroidal artery by the most likely atherothrombotic mechanism; sub intending tonic–clonic seizures	Sequelae of stroke; left hemiparesis spastic sequelae; Alzheimer’s disease; Alzheimer’s dementia; cerebrovascular disease; agenesis of the corpus callosum; cerebellar abiotrophy	Subtended tonic–clonic seizures	Confirmed infection with SARS-CoV-2.
6	66	Left transverse–sigmoid–jugular venous subacute thrombosis	Cerebrovascular disease; predominantly sensory axonal polyneuropathy; newly discovered type II diabetes; C6-C7 protrusion with left C7 radicular conflict; neurocognitive disorder	Language disorders with fluctuating evolution	Confirmed infection with SARS-CoV-2
7	71	Secondary epilepsy-generalized seizures	Secondary cerebral determinations	Two generalized tonic–clonic seizures, without sphincter relaxation, without biting tongue	Pulmonary secondary determinations; pancreatic head tumor; sequelae of ischemic stroke left posterior cerebral artery; essential hypertension; temporospatial disorientation; confirmed infection with SARS-CoV-2
8	79	Left occipital subacute stroke by atherothrombotic mechanism	Sequelae of stroke; cerebral atheromatosis; mixed dementia	Right limb motor deficit, predominantly brachial	Bilateral carotid atheromatosis; insulin-dependent type II diabetes mellitus with poor control; diabetic polyneuropathy; confirmed infection with SARS-CoV-2
9	41	Morphic seizure.	Right parietal cystic expansive process; left parietal hemangioma; chronic left parietal microhemorrhages	Headaches and seizure during his admission	Mixed dyslipidemia; confirmed infection with SARS-CoV-2
10	72	Left ischemic posterior cerebral artery stroke transformed hemorrhagic; flaccid paraplegia; exitus by cardiorespiratory arrest	Sequelae of stroke	Bilateral lower limb plegic motor deficit	Hypertension; atrial flutter; recent post-COVID-19 condition

## Data Availability

Third-party data restrictions apply to the availability of these data. The data were obtained from Constanta’s Sf Apostol Andrei County Emergency Clinical Hospital and are available from the authors with the permission of the Institutional Ethics Committee of Clinical Studies of the Constanta’s Sf. Apostol Andrei County Emergency Clinical Hospital.

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
