# Peer review of "Impact of SARS-CoV-2 Infection in Patients with Neurological Pathology"

_diagnostics, 2022, doi:10.3390/diagnostics12020473_

Round 1
Reviewer 1 Report
The manuscript entitled “Impact of SARS-CoV-2 infection in patients with neurological 2 pathology” by Axelerad et al. provided a perspective on the neurological pathology associated with the 34 SARS-CoV-2 virus. I would like to thank the authors for this interesting work. However, there are some points that should be considered to revise the manuscript: I. Recent research noticed an increasing prevalence of anxiety and circadian rhythm disorders during the COVID-19 pandemic. The authors must discuss this in the manuscript (Reference: https://doi.org/10.1007/s11356-021-18384-4). II. Clock genes regulate the circadian rhythm of biological processes in eukaryotic organisms, and their roles in neurological pathologically need to be briefly mentioned for better presentation and increasing scientific soundness (Reference: PMID: 34973458). III. COVID-19 has become a global public health obstacle. This disease has caused negligence on mental health institutions, decreased trust in the healthcare system and traditional and religious beliefs, and has created a widespread stigma on people living with mental health illnesses. Authors are encouraged to discuss this from a public health perspective to add a much-needed dimension to the article (References: https://onlinelibrary.wiley.com/doi/10.1002/hpm.3394) IV. In terms of the number of COVID-19 cases and deaths, the scales in high-income countries are much bigger than that of LMICs. The authors need to briefly discuss how the COVID-19 disease has impacted LMICs and the strain in the healthcare system caused by COVID-19. Suggested papers: https://doi.org/10.4269/ajtmh.21-1058, https://doi.org/10.1186/s41182-021-00360-w, https://doi.org/10.1016/j.cegh.2021.100841, https://doi.org/10.1017/ice.2021.257, https://doi.org/10.1002/hpm.3334 V. The authors need to ensure the English language is of sufficient quality to be understood.
Author Response
Reviewer 1
The manuscript entitled “Impact of SARS-CoV-2 infection in patients with neurological 2 pathology” by Axelerad et al. provided a perspective on the neurological pathology associated with the 34 SARS-CoV-2 virus. I would like to thank the authors for this interesting work. However, there are some points that should be considered to revise the manuscript:
- Recent research noticed an increasing prevalence of anxiety and circadian rhythm disorders during the COVID-19 pandemic. The authors must discuss this in the manuscript (Reference: https://doi.org/10.1007/s11356-021-18384-4).
We have added new information accordingly to your recommendations. LINES 1450-1461
„Sleep phase abnormalities were the most prevalent circadian rhythm abnormalities. Boiko et al. [12] discovered that COVID-19 infection in antecedents increased patients' susceptibility to developing circadian rhythm abnormalities, including delayed sleep phase disorder. Additionally, it was shown that individuals with COVID-19 exhibit el-evated levels of trait and state anxiety. The high incidence of delayed sleep phase disorder might be a result of both the neurotropic properties of SARS-CoV-2 and isolation linked with the COVID-19 outbreak. Therefore, the impact of a modification in light exposure on everyday activities was detected under lockdown settings, meaning a change of the rhythm by three or more hours. This has resulted in decreased exposure to natural light, which has a detrimental effect on the main circadian oscillator, as well as physical and nutritional activities. Amongst circadian rhythm abnormalities, delayed sleep phase disturbance is most prevalent in the post-COVID era and is related to an increased degree of anxiety in these individuals.”
- Clock genes regulate the circadian rhythm of biological processes in eukaryotic organisms, and their roles in neurological pathologically need to be briefly mentioned for better presentation and increasing scientific soundness (Reference: PMID: 34973458).
We have added new information accordingly to your recommendations. LINES 1436-1448
„Six key clock genes, including CLOCK, BMAL1, PER1, PER2, CRY1, and CRY2, control circadian rhythm. Clock genes are involved in the regulation of metabolic and immunological responses, including the release of pro-inflammatory interleukins. As a result, lifestyle modifications, including adjustments in the light regime, a reduction in the amplitude of room temperature, a change in the timing of eating, and the allo-cation of a food according to Its calorific value throughout the day, contribute to metabolic abnormalities and the apparition of a low-intensity systemic inflammatory process [11]. Additionally, the circadian clock governs fundamental bodily processes, including lung capacity and sleeping as well as activities in the neural tissue that con-tribute to neurological and psychiatric illnesses, such as auto-aggression, and neuro-pathic pain. Additionally, several studies have discovered a link between the devel-opment of certain symptoms and particular circadian chronotypes that might aid in the creation of chronotherapy and enhance therapy by administering medication in line with the patient's circadian rhythm [12, 13].”
III. COVID-19 has become a global public health obstacle. This disease has caused negligence on mental health institutions, decreased trust in the healthcare system and traditional and religious beliefs, and has created a widespread stigma on people living with mental health illnesses. Authors are encouraged to discuss this from a public health perspective to add a much-needed dimension to the article (References: https://onlinelibrary.wiley.com/doi/10.1002/hpm.3394)
We have added new information accordingly to your recommendations. LINES 1465-1529
„Formerly, it was thought that the SARS-CoV-2 virus was limited to the respiratory system, mostly impacting the lungs; nevertheless, new investigations have demonstrated the virus's multisystem impact, most notably impacting brain tissue. COVID-19 illness may indeed be associated with a variety of mental disorders, including post-traumatic stress disorder, obsessive–compulsive disorder, anxiety, delirium, and depression, either directly or indirectly. It has the potential to exacerbate pre-existing mental health problems or to precipitate the genesis of new psychiatric diseases [14]. Increasing COVID-19 instances, elevated illness burden, and lack of social support may all contribute to short-term mental health concerns, while economic losses also present because man-dated lockdowns could have a long-term effect on people's mental health [15].
Due to a global scarcity of frontline staff dedicated to containing the epidemic, several governments have reassigned psychiatrists to critical care settings to handle COVID-19 patients. As a result, mental hospital outpatient departments were closed, resulting in a massive increase in mental health difficulties in some countries. In these cases, imposed lockdowns and restrictive restrictions regarding physical isolation have robbed many patients of access to essential mental health treatment [16].
The COVID-19 pandemic has highlighted underdeveloped health systems in low- and middle-income countries (LMICs) and significant treatment disparities, at least in the area of mental health. As a result of these issues, individuals with severe mental illness die earlier, have more physical ailments, and receive less medical treatment than the general population [17].
Additionally, it is vital to educate the public in low- and middle-income countries on the acknowledgment of mental health problems as diseases, the importance of social and familial assistance, and the importance of avoiding social stigma of those who suffer from mental health disorders.”
- In terms of the number of COVID-19 cases and deaths, the scales in high-income countries are much bigger than that of LMICs. The authors need to briefly discuss how the COVID-19 disease has impacted LMICs and the strain in the healthcare system caused by COVID-19. Suggested papers: https://doi.org/10.4269/ajtmh.21-1058, https://doi.org/10.1186/s41182-021-00360-w, https://doi.org/10.1016/j.cegh.2021.100841, https://doi.org/10.1017/ice.2021.257, https://doi.org/10.1002/hpm.3334
We have added new information accordingly to your recommendations. LINES 1556-1638
„Global restrictions imposed to prevent and control the spread of a new COVID-19 wave resulted in a financial crisis in LMICs, restricting access to food and other basic requirements due to border closures. The combined consequences of poverty, climate change, and the COVID-19 epidemic have exacerbated food insecurity in some LMICs. COVID-19 thus worsened an already-existing food crisis in these countries due to the imposition of government restrictions and lockdown measures that restricted work options and income.
Food insecurity is a major socioeconomic and public health problem in low- and middle-income nations. It is associated with adverse health effects and a reduction in self-reported health status, decreased micronutrient intake, fruit, and vegetable con-sumption, overweight, and birth abnormalities. As with challenges experienced by other resource-related issues (e.g., housing instability, energy uncertainty), inequality and poverty might exacerbate nutritional deficiencies, illness, and disease management. Individuals who experience poverty have much worse health outcomes and less access to healthcare than those who do not. Poor nutrition may exacerbate pre-existing illnesses, such as inadequate glucose control in diabetic patients, final renal disease in patients with chronic kidney disease as well as affecting treatment of other chronic diseases. COVID-19's presence in LMICs has restricted access to healthcare and impacted attempts to treat, diagnose, immunize, and monitor other infectious diseases. Food instability may further aggravate health problems and expenditures for families with children that have specific healthcare requirements, or for persons with disabilities.
A number of examples can be provided depicting the impact of the COVID-19 pandemic on LMICs.
Afghanistan's continuous struggle has created several difficulties for the country's population. Afghanistan has seen a significant rise in food shortfalls because of its reliance on neighboring nations during the epidemic [18]. Individuals attain food security when they have continuous physical and economic access to an adequate supply of safe and nutritious food that meets their dietary demands and preferences. Food scarcity, political unrest, and the third wave of COVID-19 have made it impossible to obtain basic supplies. Consequently, folks are forced to contend with the COVID-19 pandemic amid economic collapse and despair. At this crucial point, worldwide efforts are essential to ameliorate food security.
The growth in the number of instances of the illness has the potential to overwhelm the health system, as well as noncompliance with social distance measures and the in-troduction of variations of concern in LMICs. This rise in the transmission curve may also create conditions favorable for the emergence of further changes in the virus' structure and DNA. As a result, genomic monitoring methods are essential to detect and describe these variations as well as to determine if the vaccines against the virus that are currently in use are efficacious.
The development of efficient and dependable infectious disease monitoring systems is critical for establishing a high-quality public healthcare system and reducing the mortality rates in low- and middle-income countries. Monitoring helps facilitate accessi-bility of records and knowledge and decreases the burden and propagation of unfa-vorable healthcare events. This allows for a rapid response in public health, efficient implementation of methods and countermeasures, and review of suggested treatments, rapid detection of new illnesses, and health security and stability for people living in LMICs.
Healthcare workers (HCWs) have been critical in containing the pandemic and mitigating its effects. Increased working hours and frequent exposure to critically ill patients have major consequences for of the health and wellbeing of physicians, which have previously been disregarded.
Infectious disease epidemics have always posed a threat to public health, especially in Africa, where outbreaks have exploded in recent years [19]. Although several infec-tious diseases have emerged in Africa, such as Ebola and certain other epidemic-prone infections, insufficient focus has been placed on the development of health surveillance systems. The inadequacy of the region’s healthcare monitoring techniques have only recently been identified. Africa suffers from a shifting epidemiology of disease, a deficient healthcare system, and a scarcity of resources. Only a vigilant monitoring system ensure that the most use of its available resources are made in an effective and strategically managed manner. Measures are needed to rapidly detect potential public health threats. This could be accomplished via the use of appropriate, efficient, and lengthy surveillance methods.
Dengue fever is a serious public health concern in Africa [19] and the COVID-19 pandemic has exacerbated this concern. COVID-19 accelerates the transmission of a va-riety of illnesses, including zika, yellow fever, measles, mucormycosis, Lassa fever, and HIV, as has been seen in several nations on the African continent. The limits imposed in response to the COVID-19 pandemic have resulted in the suspension of vector man-agement initiatives that aid in the management of these diseases. To avert further public health disaster, urgent and interdisciplinary measures to dengue fever epidemic man-agement in African nations are necessary.
COVID-19 cases have impoverished Nigeria’s healthcare system and resulted in additional neglect of persons suffering with mental illness [19]. In general, there is a demand for equitable access to healthcare resources, but there is a need for adequate attention and treatment for mental health patients, which is rising in Nigeria.
The COVID-19 epidemic has hit public health emergencies in Bangladesh, a low–middle-income nation in South Asia. The surge of sickness cases since may generate overburdening of the health system. This rise in the contagion curve may also encourage further alterations in the virus' structure and DNA. It is important to find, monitor, and characterize these polymorphisms and determine the efficacy of existing vaccinations against these variants.
India’s healthcare sector is suffering significant difficulties as a consequence of the lack of resources to combat the COVID-19 pandemic, with HCWs also suffering the consequences [20]. There is an urgent need to address these flaws in the healthcare system to provide a consistent and ongoing supply of high-quality treatment in India.”
- The authors need to ensure the English language is of sufficient quality to be understood.
Also, we have English-edited the article in the MDPI service. We hope that this revised version of the article will provide new, insightful, and relevant information.
Thank you for your help, guidance, and support!
Reviewer 2 Report
This paper is a case report with 10 patients with COVID and neurological diseases.
The authors did not categorize or clarify the diagnostic features for the impact of SARC-CoV-2 on neurological diseases. Thus, I wonder what the authors found? What is new? They should summarize diagnostic findings as one table or more.
To avoid the limitation of care reports, they should focus on summary what they found.
Minor, the figure legends and table should be revised to explain each figure (what is a,b,,,f?) and to merge the tables into one.
Author Response
Reviewer 2
This paper is a case report with 10 patients with COVID and neurological diseases.
The authors did not categorize or clarify the diagnostic features for the impact of SARC-CoV-2 on neurological diseases. Thus, I wonder what the authors found? What is new? They should summarize diagnostic findings as one table or more.
We have categorize the diagnostic features and symptoms in the table below.
We have added new information accordingly to your recommendations. LINES 1293-1312
„The COVID-19 patients in our 10 case reports presented with a complex panel of neurological diagnostics, including myelitis with paraparesis, acute ischemic stroke in various territories with hemiplegia, cerebrovascular disease with involuntary move-ments, seizures, and left transverse–sigmoid–jugular venous subacute thrombosis.
Cerebrovascular disease, in addition to certain other neurological features, has often been associated with acute SARS-CoV2 infection. Numerous pathophysiological mecha-nisms have been postulated to explain the SARS-CoV2-related prothrombotic condition, as both direct and indirect consequences of the viral infection. Aside from hypercoagulable characteristics, it is hypothesized that SARS-CoV2-related endothelitis and microangi-opathy lead to hemorrhagic stroke. Consequently, intracranial hemorrhage in COVID-19 patients could be the result of hemorrhagic transformation of ischemic stroke, original hemorrhagic stroke, or traumatic intracranial hemorrhage.
The processes underlying the apparition of these neurological symptoms remain unknown. Numerous ideas have been advanced since SARS-CoV-2 was first detected, such as that the neuroinvasion of the virus comes from its ability to enter via the olfactory groove or directly into the nervous system via circulation [6-8]. However, these results might be the result of secondary immunological processes and a severe inflammatory state induced by infection, or of significant hypoxia caused by critical illness and con-comitant disorders [6-8].”
Case ID |
Age |
Neurological features |
Other diagnostics |
||
|
|
Neurological-COVID-19-related diagnostic |
Other neurological diagnoses |
Neurological symptoms |
|
1 |
49 |
Myelitis; paraparesis |
|
Ascending lower limb paresthesia and lower limb motor deficit |
Biological inflammatory syndrome; acute urinary retention; confirmed infection with SARS-CoV-2. |
2 |
78 |
Acute ischemic stroke in the superficial and deep left sylvian territory and in the superficial border territories, middle cerebral artery/posterior cerebral artery and middle cerebral artery/left anterior cerebral artery by cardioembolic mechanism; right hemiplegia |
|
A crisis of loss of consciousness, language disorder and right limb motor deficit |
Permanent atrial fibrillation; hypertension; gout; biological inflammatory syndrome; interstitial pneumonia; confirmed infection with SARS-CoV-2. |
3 |
57 |
Cerebrovascular disease; involuntary right upper limb movements |
Cerebral atherosclerosis |
Confusion syndrome and involuntary right upper limb movements |
Dilated cardiomyopathy with left ventricular dysfunction; congestive heart failure class III (New York Heart Association (NYHA)); mild mitral regurgitation; acute chronic kidney disease; upper rectal neoplasm operated and radio chemically treated; hypertension; confirmed infection with SARS-CoV-2. |
4 |
69 |
Cerebrovascular disease |
Sequelae of stroke; right sequelae hemiparesis |
Right limb motor deficit and speech impairment with onset |
Left internal carotid artery occlusion; hypertension; type II diabetes mellitus with diabetic nephropathy. Chronic kidney disease grade 3B; left thigh amputation; confirmed infection with SARS-CoV-2. |
5 |
68 |
Subacute ischemic stroke in the territory of the anterior choroidal artery by the most likely atherothrombotic mechanism; sub intending tonic-clonic seizures |
Sequelae of stroke; left hemiparesis spastic sequelae; Alzheimer’s disease; Alzheimer’s dementia; cerebrovascular disease; agenesis of the corpus callosum; cerebellar abiotrophy |
Subtended tonic clonic seizures |
Confirmed infection with SARS-CoV-2. |
6 |
66 |
Left transverse–sigmoid–jugular venous subacute thrombosis |
Cerebrovascular disease; predominantly sensory axonal polyneuropathy; newly discovered type II diabetes; C6-C7 protrusion with left C7 radicular conflict; neurocognitive disorder |
Language disorders with fluctuating evolution |
Confirmed infection with SARS-CoV-2 |
7 |
71 |
Secondary epilepsy-generalized seizures |
Secondary cerebral determinations |
2 generalized tonic-clonic seizures, without sphincter relaxation, without biting tongue |
Pulmonary secondary determinations; pancreatic head tumor; Sequelae of ischemic stroke left posterior cerebral artery; essential hypertension; temporospatial disorientation; confirmed infection with SARS-CoV-2 |
8 |
79 |
Left occipital subacute stroke by atherothrombotic mechanism |
Sequelae of stroke; cerebral atheromatosis; mixed dementia |
Right limb motor deficit predominantly brachial |
Bilateral carotid atheromatosis; insulin-dependent type II diabetes mellitus with poor control; diabetic polyneuropathy; confirmed infection with SARS-CoV-2 |
9 |
41 |
Morphic seizure. |
Right parietal cystic expansive process; left parietal hemangioma; chronic left parietal microhemorrhages |
Headaches and seizure during his admission |
Mixed dyslipidemia; confirmed infection with SARS-CoV-2 |
10 |
72 |
Left ischemic posterior cerebral artery stroke transformed hemorrhagic; flaccid paraplegia; exitus by cardiorespiratory arrest |
Sequelae of stroke |
Bilateral lower limb plegic motor deficit |
Hypertension; atrial flutter; recent post-COVID-19 condition |
To avoid the limitation of care reports, they should focus on summary what they found. Minor, the figure legends and table should be revised to explain each figure (what is a,b,,,f?) and to merge the tables into one.
The tables were merged into one and the legends for the figures were updated. Also, we have English-edited the article in the MDPI service. We hope that this revised version of the article will provide new, insightful, and relevant information.
Thank you for your help, guidance, and support!
This manuscript is a resubmission of an earlier submission. The following is a list of the peer review reports and author responses from that submission.
Round 1
Reviewer 1 Report
Through case reports, this study demonstrates that covid-19 has an effect on the nervous system. This is a descriptive study. It is best to sort out and summarize the study through some quantitative indicators. The manuscript is not enough to analyze the impact of covid-19 on the nervous system, especially the mechanism. Covid-19 patients firstly show gastrointestinal symptoms rather than respiratory symptoms. Whether the neurological complications caused by covid-19 are caused by vagal nerve mediated intestinal brain axis needs to be emphasized.
Reviewer 2 Report
The authors performed a study to describe the neurologic features of patients with confirmed COVID-19.
The study would be interesting, however, it has several problems in the methodology description, results presentation and discussion.
Overall, the manuscript looks like a draft, rather than the final version. The text has many errors in the English language, with several sentences written in an atypical way, which makes the text difficult to understand.
For example in the legend of Figure 1:
“Figure 1. Inaccurately delimited discrete hyperintense foreshores T2-STIR/hypointense T1, without any notable contrast outlet, disposed at the level of lateral coordinates of the marrow corresponding to T6-T10 myeloma; -Thoracic intramedullary lesions with suggestive characteristics for an inflammatory-infectious sublayer.” What did the authors mean by "marrow"? Was it spinal cord?
What did the authors mean by "contrast outlet"? Was it contrast-enhancement?
Other example: "Figure 10. Pulmonary lesions with polymorphic aspects, some in the form of inaccurately delimited areas of matt-glass clouding..." What did the authors mean by "matt-glass"? Was it ground glass opacities?
These are just 3 examples of errors in imaging or physical examination findings descriptions that make the text confusing and scientifically unreliable.
Material and methods:
The study methodology was not described. The authors did not describe the inclusion and exclusion criteria, what parameters they considered in the study for the enrolled patients, and for the final sample.
For example:
Were the RT-PCR tests for SARS-CoV-2 performed only in the oropharynx or nasopharynx? Did any patient performed the RT-PCR tests for SARS-CoV-2 in the cerebrospinal fluid?
Results:
The case-by-case description of 10 patients is confusing and can mislead the reader.
In addition, in some cases, the neurological alterations seem to be unrelated with the COVID-19, such as in cases 4 and 7.
Also, some cases are not adequately described. For example, I am not sure if the brain MRI alterations described in case 3 are demyelinating lesions.
A important flaw in the results is that the hospital where the study was carried out does not seem to be a reference for the treatment of CVID-19, and the patients were discharged for other hospital for treatment. Therefore, the description of the patients' clinical evolution is not described, making the study incomplete.
Discussion:
The study is not well discussed. The authors did not include several important studies that reported neurological changes related to COVID-19.
Also, a limitation paragraph is lacking.